# D^2^BOF-COVIDNet: A Framework of Deep Bayesian Optimization and Fusion-Assisted Optimal Deep Features for COVID-19 Classification Using Chest X-ray and MRI Scans

**DOI:** 10.3390/diagnostics13010101

**Published:** 2022-12-29

**Authors:** Ameer Hamza, Muhammad Attique Khan, Majed Alhaisoni, Abdullah Al Hejaili, Khalid Adel Shaban, Shtwai Alsubai, Areej Alasiry, Mehrez Marzougui

**Affiliations:** 1Department of Computer Science, HITEC University, Taxila 47080, Pakistan; 2Computer Sciences Department, College of Computer and Information Sciences, Princess Nourah bint Abdulrahman University, Riyadh 11671, Saudi Arabia; 3Faculty of Computers & Information Technology, Computer Science Department, University of Tabuk, Tabuk 71491, Saudi Arabia; 4Computer Science Department, College of Computing and Informatics, Saudi Electronic University, Riyadh 11673, Saudi Arabia; 5College of Computer Engineering and Sciences, Prince Sattam bin Abdulaziz University, Al-Kharj 16278, Saudi Arabia; 6College of Computer Science, King Khalid University, Abha 61413, Saudi Arabia

**Keywords:** COVID-19, deep learning, Bayesian optimization, fusion, optimization, neural network

## Abstract

Background and Objective: In 2019, a corona virus disease (COVID-19) was detected in China that affected millions of people around the world. On 11 March 2020, the WHO declared this disease a pandemic. Currently, more than 200 countries in the world have been affected by this disease. The manual diagnosis of this disease using chest X-ray (CXR) images and magnetic resonance imaging (MRI) is time consuming and always requires an expert person; therefore, researchers introduced several computerized techniques using computer vision methods. The recent computerized techniques face some challenges, such as low contrast CTX images, the manual initialization of hyperparameters, and redundant features that mislead the classification accuracy. Methods: In this paper, we proposed a novel framework for COVID-19 classification using deep Bayesian optimization and improved canonical correlation analysis (ICCA). In this proposed framework, we initially performed data augmentation for better training of the selected deep models. After that, two pre-trained deep models were employed (ResNet50 and InceptionV3) and trained using transfer learning. The hyperparameters of both models were initialized through Bayesian optimization. Both trained models were utilized for feature extractions and fused using an ICCA-based approach. The fused features were further optimized using an improved tree growth optimization algorithm that finally was classified using a neural network classifier. Results: The experimental process was conducted on five publically available datasets and achieved an accuracy of 99.6, 98.5, 99.9, 99.5, and 100%. Conclusion: The comparison with recent methods and t-test-based analysis showed the significance of this proposed framework.

## 1. Introduction

In December of 2019, the consumption of bat meat at an unusual animal meat market in Wuhan, Hubei, China was connected to a group of people contracting pneumonia of an unknown origin [1]. The pandemic soon spread to other regions of the globe, and on 11 March 2020, the World Health Organization declared COVID-19 a worldwide pandemic outbreak that continues to this day. A new type of beta coronavirus was found using the unbiased sequencing of patient samples. The 2019-nCoV coronavirus, in comparison to the Middle East respiratory syndrome (MERS) and the severe acute respiratory syndrome (SARS), has a higher transmission potential and a lower mortality rate; SARS and MERS are both animal-borne diseases, and civets and camels, respectively, were known to carry the diseases [2]. The emergence of diseases like SARS and MERS, both of which are suspected to have been brought on by new coronaviruses, is more possible when there are no borders between human civilization and the natural environment. In the early days, the number of deaths continued because there was no vaccination [3].

Research into different early detection techniques was essential to combating this pandemic epidemic [4]. To date, according to the updated figures from the WHO, 6.49 million fatalities have occurred all over the world, and 30,581 have been from Pakistan, while 104,000,000 have been Americans [5]. Reverse transcription polymerase chain reaction is presently the gold standard for detecting coronavirus infections (RT-PCR). Using a nasopharyngeal swab, the RT-PCR technique identifies viral RNA. The most significant limitation of the RT-PCR test is its limited sensitivity range. As a result, the RT-PCR test is useless for quickly determining the rate of positive instances. To resolve this restriction, imaging-based biomedical methods, including computed tomography (CT) images, radiography, and chest X-rays, are cheap methods for the diagnosis of COVID-19 [6]. It is possible to stop the spread of a coronavirus infection if biomedical imaging is utilized to detect an infection in its initial phases [7]. Methods for automatically classifying COVID-19 images that typically include preprocessing, feature extraction, and selection have been demonstrated to be better than human COVID-19 classification techniques. As a direct result, it is important to create a decision assistance system that uses artificial intelligence (AI) [8]. The AI method uses images to distinguish the illness apart from others and identify it at the lung level. In COVID-19 patients, CT scans help locate organ damage and classify its development. This is because some COVID-19 patients contract lung infections [9]. The radiological imaging of COVID-19 patients may reflect that of individuals with bacterial or viral pneumonia, especially SARS and MERS-associated pneumonia. Consequently, the ability to effectively differentiate diseases via the analysis of medical images has arisen as a significant challenge, and its resolution requires supporting health personnel with early disease diagnosis, as well as the rapid separation of infected patients [10].

The development of machine learning approaches that might help in the diagnosis of COVID-19 using chest X-ray or CT scan images is the subject of several initiatives in the area of medical image processing. CT scans have disadvantages over CXR, including restricted access to equipment, radiologists, physicians, higher prices, and longer image-acquisition periods [11]. Usually, the handcrafted features are extracted from the input images and passed to the ML method for classification [12]. The handcrafted features include texture, shape, and point-based information of the image. Recently, the entrance of deep learning has shown huge success in the area of medical imaging [13]. Deep learning is a type of machine learning that need a considerable amount of data for training a model. It enables the development of computer models with many processing layers that can learn how to represent data at various levels of complexity. This technology facilitates the development of recognition-based applications, such as object detection, speech recognition [14], and medical imaging [15].

The medical images of COVID-19 patients may reveal a pattern based on the presence of common traits [16]. Deep learning is a useful approach that researchers employ to help medical practitioners comprehend huge amounts of data, such as chest X-ray scans [17]. However, some challenges that are faced in the deep learning research for COVID-19 include: (i) inadequate amounts of training data; (ii) manual initialization of the hyperparameters of the deep model; and (iii) the addition of some irrelevant features that degrade the accuracy of a proposed framework. In this work, we proposed an automated optimized Bayesian deep learning-based framework for COVID-19 classification, as shown in Figure 1. The major contributions of the proposed framework are listed as follows:Prepared a large dataset using flip and rotation operation from the original dataset and trained pre-trained deep learning models;Opted for the freezing weights procedure and selected hyperparameters through Bayesian optimization for efficient training using selected datasets;Proposed an improved canonical correlation analysis fusion technique for feature fusion of both deep learning models;Proposed an improved tree growth optimization for best feature selection.

The rest of the manuscript is organized in the following order. Related works are discussed under Section 2, which includes the recent studies and their brief summaries, such as cutting-edge work and gaps in the research. The proposed framework is presented under Section 3, which includes the detailed mathematical formulation and visual facts. The results are presented under Section 4. Finally, Section 5 concludes the paper.

## 2. Literature Review

In earlier years, computer vision researchers developed many algorithms for the identification and classification of COVID-19 using CXR images [18,19]. Few researchers developed innovative deep learning (DL) architectures for the detection and identification of corona viruses from CXR and CT images, while the majority of studies focused on traditional techniques [20,21]. Muhammad et al. [16] presented a framework for corona virus classification from CXR images using deep explainable AI. For the training and feature extraction processes, two deep learning models were used. They used canonical correlation analysis to improve feature fusion. Furthermore, the hybrid whale-elephant herding feature section was used to optimize fused features. Three publicly available datasets were used by the authors. They achieved accuracies of 99.1, 98.2, and 96.7%, which were better than previous techniques. The limitation of this work was the optimization algorithm’s static threshold value, which will be resolved in future work. Ameer et al. [22] presented a framework by employing CNN-LSTM for corona classification using CXR images. They developed a novel CNN-LSTM method with modified EfficientNetB0 for deep feature extractions. Additionally, extracted features were fused using serial-based maximum value fusion techniques, and improved moth flame feature selection was employed on the fused vector. The studies were carried out on three publicly available datasets and yielded accuracy rates of 93.0, 94.5, and 98.5%, respectively. The drawback of this work was the fusion process, which controlled the vector size and increase the computational time. Xiaole et al. [23] introduced a novel branch model network using transformations and CNN for the recognition of CT scan images. They implemented two branches-based models. One was built using CNN and the second one using transformation-based branches. The features were fused the using bi-directional approach. They used a large scale COVID-19 dataset for the experiment and achieved a 96.7% accuracy rate. The limitation of this research was the incomplete information of patients and the inadequate amount of features. Aksh et al. [24] implemented an efficient CNN model for the detection of COVID-19 using CXR and CT images. They designed a CNN model that included several layers and visualized weights through GradCam visualization. The entire introduced method was implemented on the COVID-19 multiclass CT dataset and achieved a 97.6% accuracy rate. The drawback of this research was the inadequate amounts of data used for the training process. Gayathri et al. [25] created a computer-aided mechanism for the diagnosis of COVID-19 via CXR images. The presented method was based on the DCNN and sparse auto encoder. The experiments were performed on COVID-19 and non-COVID images and attained an accuracy of 95.7%. AbdElhamid et al. [26] developed a COVID-19 multi-classification technique using CXR images. An XceptionNet pre-trained model that was trained using TL, and obtained features from the GAP layer were part of the proposed model. A three-class, open source dataset was used in the experiment, which had a 99.3% accuracy rate. The limitation of this method was the inadequate number of images used in the selected dataset. Rahul et al. [27] employed a framework that diagnosed COVID-19 using deep features and correlation coefficients. In that study, the authors applied DCNN for feature extraction that was further utilized for the classification. Veerraju et al. [28] implemented a novel technique that diagnosed COVID-19 by adapting hyperparameters via a hosted cuckoo optimization algorithm. Samritika et al. [29] designed an automated detection and classification framework using chest images via CNN. The authors performed two classification tasks—binary and multiclass—and achieved improved accuracy. Vruddhi et al. [30] presented a method of diagnosing COVID-19 using CT images and deep learning methods. They designed a novel CNN model named CTnet-10. The selected dataset consisted of two classes, and the experiments attained a higher accuracy of 82.1%. Moreover, they used traditional DCNN networks and achieved a 94.52% accuracy rate. Umut et al. [31] presented an automated and effective method for the detection of coronavirus disease. The authors extracted the features using four CNN architectures and fused the information using the ranking-based technique. The proposed method achieved a 98.93% accuracy rate. The disadvantage of this work was the ranking-based fusion, because it missed the important features. Ghulam et al. [32] presented a multi-layer fusion for the classification of coronavirus disease from lung ultrasound images. The presented model was designed by five main blocks of convolutional connectors and employed the fusion technique. The open source dataset was selected for the experimental process and achieved a 92.5% accuracy rate. The high number of parameters was the major limitation of this work. Emtiaz et al. [33] presented a deep learning-based classification framework using CXR images. The authors designed novel CNN architecture based on 22 layers that were further employed for the classification. For a binary dataset, the presented framework obtained 99.1% accuracy, whereas the multiclass accuracy was 94.2%. Dalia et al. [34] presented an optimized deep learning network using the GSO algorithm. The employed approach was performed on the binary class dataset. By this approach, they achieved a significant accuracy of 98%. Abirami et al. [35] presented a novel framework based on generative adversarial networks for the classification of COVID-19 using medial CXR images. The augmentation process was employed by using GAN, and the generated samples fed to a novel created network. The described framework achieved 99.78% accuracy. Abirami et al. [36] presented a framework that automated segments and identified COVID-19 lung infection using CT-scan images. The created model achieved 98.10% accuracy for classification and the segmentation achieved 81.1% accuracy for the dice coefficient using GAN segmentation. Irfan et al. [37] presented an automated framework for diagnosing the COVID-19 disease using X-ray images. The models Densenet121, Resnet50, VGG16, and VGG19 were trained using transfer learning. The CXR and normal images were collected from four different publically available datasets. The dataset consisted of two classes (COVID and normal). Using this approach, the presented framework achieved 99.3% accuracy. VGG16 and VGG19 outperformed the other two models. The limitation of this work was that the authors only collected the COVID-19 and normal images from the different datasets. The authors removed other classes, such as pneumonia. The presented framework was unable to diagnose the other respiratory infections. Naeem et al. [38] presented a novel method that detected the infection of the COVID-19 disease by using chest radiography images. The described model had nine convolutional and one fully connected layer. The provided architecture used two activation functions: the ReLu activation function and the Leaky Relu. The model experiments were conducted on multiclass datasets. The datasets consisted of three classes (COVID-19, normal, and pneumonia). Using this approach, the authors achieved 98.40% accuracy. Shifat et al. [39] described a technique based on a Bayesian optimization of deep learning approach for the classification of the COVID-19 disease using X-ray images. The presented framework developed a novel DCNN model named COVIDXception-Net and it was trained by employing Bayesian optimization for the selection of the best-trained model. The authors performed the whole experiment on four publically available datasets, and the provided framework achieved 99.2% of accuracy. At the end, they performed qualitative analysis by utilizing the GRAD-CAM visualization.

In the summary, the authors in the literature used pre-trained models with transfer learning concepts for COVID-19 classification. Few of them focused on the binary class problem, and many of them considered the multiclass problem. The deep models were trained using static hyperparameters, such as learning rate, depth section momentum, and the number of epochs. In addition, the authors selected a relatively small number of datasets for the training process. There are several challenges in effectively classifying COVID-19 using standard chest X-rays. Individuals with COVID-19 may have radiological imaging that resembles that of patients with bacterial or viral pneumonia, most notably those caused by SARS and MERS. As a result, the ability to correctly diagnose diseases by examining medical imagery has become a critical challenge. The first issue is the classification of multiple classifications, including COVID-19, viral pneumonia, lung opacity, TB, fibrosis patterns, and normal images. This is a challenge, since there are so many different types of lung diseases. These images are shown quite well in Figure 2. This figure illustrates that there is a high degree of resemblance between each image, which means that there is a chance that an incorrect classification will be made. The second challenge is the removal of redundant and useless information, which lowers the accuracy of classification, while simultaneously increasing computation time.

## 3. Proposed Methodology

This section explains the proposed COVID-19 classification method using DCNN via Bayesian optimization and an improved tree growth feature selection with detailed visual illustrations and mathematical equations. Figure 1 depicts the proposed COVID-19 classification framework. This diagram shows that the publicly available datasets were obtained, and data augmentation was performed at the first step. Following that, two pre-trained deep models, ResNet50 and Inception V3, were used and modified based on the nature of the selected datasets. Transfer learning (TL) was used to train both models, and Bayesian optimization was used to initialize the hyperparameters. Features were extracted from both trained models and fused using improved canonical correlation analysis that used the activation function. Improved tree growth optimization was used to further optimize the fused features. The best features were then fed into neural network classifiers for final classification accuracy. Below is a brief description of each step.

### 3.1. Dataset Selection and Normalization

In this work, we used five publically available datasets for the experimental process. The selected datasets are Chest X-ray (https://www.kaggle.com/datasets/prashant268/chest-xray-covid19-pneumonia (accessed on 27 November 2022)), COVID-19 Patients Lungs X-ray Images (https://www.kaggle.com/datasets/nabeelsajid917/covid-19-x-ray-10000-images (accessed on 27 November 2022)), COVID-19 Lung CT Scans (https://www.kaggle.com/datasets/luisblanche/covidct (accessed on 27 November 2022)), COVID-19 Detection (https://www.kaggle.com/datasets/donjon00/covid19-detection (accessed on 27 November 2022)), and COVID-19 Image dataset (https://www.kaggle.com/datasets/pranavraikokte/covid19-image-dataset (accessed on 27 November 2022)). The three classes in the Chest X-ray dataset include COVID-19, normal, and pneumonia. COVID-19 Patients Lungs X-ray Images dataset has two classes, which are COVID-19 and normal. The COVID-19 Lung CT Scans dataset consists of CT images. It has two classes: COVID-19 and non-COVID-19. COVID-19, normal, pneumonia, tuberculosis, and fibrosis are the five classes in the COVID-19 Detection dataset. In the COVID-19 Image dataset, three classes exist: COVID-19, normal, and viral pneumonia. A few sample images are shown in Figure 2. The images in these datasets were not enough for training, as is shown in Table 1 (training images column). Therefore, we performed data augmentation based on three operations: flip right, flip left, and rotate 90 degrees. These operations were performed for each class for all five selected datasets. The number of target images for each class included 4000; therefore, we performed these operations multiple times. The details of the datasets are shown in Table 1. Moreover, these operations are visually shown in Figure 3.

Consider the input image denoted by *f*_(*x*,*y*)_ with dimension *r* × *c*, where *x* represents the row pixels {x∈(1,2,3,…,r)} and *y* represents the columns pixel values {y∈(1,2,3,…,c)}, respectively. On input image *f*_(*x*,*y*)_, three operations are performed to augment the data. The performed operations are flip right, flip left, and rotate 90 degrees.
(1)$$f_{\left(x,y\right)}^{Right}=f_{x}\left(c+1-y\right)$$
(2)$$f_{\left(x,y\right)}^{Left}=f_{y}\left(m+1-x\right)$$
(3)$$f_{\left(x,y\right)}^{rot90}=\left[\begin{array}{cc} \cos90 & -\sin90\\ \sin90 & \cos90 \end{array}\right]\left[\begin{array}{c} f_{x}\\ f_{y} \end{array}\right]$$
where $f_{\left(x,y\right)}^{Right}$ denotes the flip right, $f_{\left(x,y\right)}^{Left}$ denotes the flip left, and $f_{\left(x,y\right)}^{rot90}$ represents rotate 90 degrees, respectively. This process was performed on all the classes of selected datasets, which had an inadequate amount of images.

### 3.2. Updated Pre-Trained RestNet50

ResNet-50 is a convolutional neural network with 50 layers. ResNet, which stands for residual networks, is a neural network type that facilitates the development of several computer vision applications [40]. Its most significant breakthrough was its capacity to train neural networks with over 150 layers. Deep CNN networks have several difficulties, including degradation concerns, gradient vanishing problems, and network optimization difficulties. They also facilitate complex jobs and enhance detection precision. ResNet seeks to address the saturation and precision loss issues that emerge during CNN training [41]. In this paper, we used the ResNet50 model for feature extraction. The model was originally trained with the ImageNet dataset, which includes 1000 object classes. The weights of the first 110 layers of the model were frozen. The FC layer, softmax layer, and classification layer were replaced with a new FC Layer, softmax layer, and a new classification layer to achieve transfer learning. We trained the updated model by using Bayesian optimization on selected datasets. The explanation of BO is provided in Section 3.4. The process of freezing weights and updating the ResNet50 architecture is visually shown in Figure 4.

### 3.3. Updated Pre-Trained InceptionV3

The GoogleLeNet network, which is similar to CNN, was created in 2014 [42]. It makes use of the inception network design, which increases overall node count while reducing a total number of network parameters [43]. GoogleNet is sometimes referred to as the Inception Network in acknowledgment of the inception network acting as the network’s skeleton. Inception v1 (2014), Inception v2 (2015), Inception v3 (2015), Inception v4 (2016), and Inception-ResNet are the most widely used GoogleNet versions (2016). In contrast to Inception v1 and v2, Inception v3 uses a convolution kernel splitting technique to divide large volume integrals into tiny convolutions. For instance, 33 convolutions are split into 31 and 13 convolutions. By minimizing the number of parameters, a splitting strategy may speed up network training and improve the retrieval of spatial data. The network structure module’s usefulness is improved in Inception v3 by using three different grid sizes (35 × 35, 17 × 17, and 8 × 8) [44]. In this research, we have utilized the Inception v3 model for deep feature extraction. The original model was trained on 1000 classes, and the size of the input layer was 299 × 299 × 3. The original network was updated by freezing the weights of 110 additional layers, and modifying the FC, softmax, and classification layers for achieving the TL. The updated model was trained using BO. The hyperparameters were selected via Bayesian optimization, as is described in Section 3.4. Visually, the process of freezing weights via the transfer learning of Inception v3 is presented in Figure 5.

### 3.4. Hyperparameters Tuning Using BO

Bayesian optimization (BO) is a method for optimizing the parameters of any black-box function *f*(*y*). Black-box optimization issues employing a black-box objective function *f* are a part of deep learning optimization (*y*). When reducing the number of layers, it is crucial to decrease the number of samples used for each step. Bayesian optimization is highly useful when human expertise cannot improve accuracy in a particular sector. By using previous knowledge about the function *f* and updating posterior information, Bayesian optimization reduces loss and increases model accuracy. Since hand tuning is based on trial and error, it is difficult to replicate. Although a grid search is not scalable to bigger dimensions, the enhanced random search is similar to the greedy method in that it settles for local optima and is unable to identify global optima. Other evolutionary optimization techniques need a lot of training cycles and are noisy. All of these problems are solved using Bayesian optimization, which properly reveals the global optimum’s black-box function. It achieves global minima by using discontinuous sections and successfully handling noisy data [45]. The optimization is achieved using the Bayes theorem, where the equation is derived as the model *M* and observation *D*.
(4)$$P\left(M|D\right)=\frac{(P(D|M)\times P\left(M\right)}{P\left(M\right)}$$
where (P(D|M) is the posterior probability of given model *M* and observation *Y*. The (P(D|M) increases the chance of observation *D* of given model *M*, and *P*(*M*) is the marginal probability of *M*. To identify the minimal value of a function *f*(*y*) on a limited set *D*, the Bayesian optimization is utilized. It is allowed to choose better decisions when further evaluations of the function *f*(*y*) are performed.

Hyperparameters are a set from variables that are employed in training and testing to speed up the learning process. The deep learning model will use a range of samples and weights to learn many feature combinations as patterns. Examples of hyperparameters include learning rate, iterations, hidden layers, batch size, activation functions, momentum, and regularization. For image classification, convolutional neural networks consider the pooling and convolutional layer fields, as well as the stride parameter-controlled step size. The parameters may range between the upper and lower limits and can be discrete, continuous, or categorical variables. In intermediate layers, the number of neurons may vary between layers, which may result in the formation of new hyperparameters. Before training the model, consider the hyperparameters p1,  p2, and p3 in the configuration space *P*, where *p* may be initialized as P=p1×p2×p3. To properly train the model, the appropriate hyperparameters should be set during the training process, since this may have a substantial impact on the model. If the model learning rate is slower, it may accidentally overlook important patterns. The right combination of parameters should improve the model efficiency or accuracy while minimizing its loss function. Thus, the topic of modifying hyperparameters may be regarded as an optimization issue [46]. In our work, we have used the Bayesian optimization for optimizing the hyperparameters of the DCNNs for improving the efficiency and minimizing the loss factor. We considered the following hyperparameters (HP): learning rate, section depth, momentum, L2Regularization, dropout rate, and activation function type. The ranges of these hyperparameters is given under Table 2.

Features are extracted from both trained models using some specific layers, such as fully connected or average pooling. For the newly trained ResNet50 deep model, the average pooling layer is selected and performs the activation function. On this layer, we obtained 2048 features; hence, the required vector size was of the dimension *N* × 2048 Similarly for the newly trained Inception v3 deep model, features were extracted from the GAP layer, and 2048 features were obtained for each image. Hence, the resultant vector was obtained of dimension *N* × 2048. In the later step, features were fused using an ICCA-based approach.

### 3.5. Improved Canonical Correlation Analysis Fusion

Feature fusion is a technique for combining many features into a single vector. The fusion technique significantly improves the performance of pattern recognition applications due to the unique properties of each feature type. When all features are merged, then there is a chance that performance will be improved. However, duplicate information might appear if many attributes are combined into one vector. In this research, we employed improved canonical correlation analysis (ICCA) for feature fusion. In the ICCA fusion, we first fused the training and testing features of both models using a serial-based approach. To resolve the high dimension and redundant problem, the features were further passed to the CCA process.

Consider the training and testing features $V_{N}^{f1},V_{N}^{f2},V_{N}^{f3},V_{N}^{f4}$ having size *N* × 2048, which is obtained from the trained Resnet50 and Inception v3. Suppose $\;V_{fused}^{train}$ and $\;V_{fused}^{test}$ are fused vectors having a size of *N* × *F*.
(5)$$\;V_{fused}^{train}={\left(\begin{array}{c} V_{N}^{f1}\\ V_{N}^{f3} \end{array}\right)}_{N\times F}$$
(6)$$\;V_{fused}^{test}={\left(\begin{array}{c} V_{N}^{f2}\\ V_{N}^{f4} \end{array}\right)}_{N\times F}$$
where $\;V_{fused}^{train}$ and $\;V_{fused}^{test}$ represent the serial-based fusion of training and testing features of both models, respectively. The sizes of both fused vectors are *N* × 4096. After that, both fused vectors are utilized for the improved canonical correlation analysis of features. Consider, ${\left\{\;V_{fused}^{train}\right\}}_{i=1}^{n}\in {ℛ}^{k}$ and ${\left\{\;V_{fused}^{test}\right\}}_{i=1}^{n}\in {ℛ}^{m}$, where *k*, *m* denotes the size of sample space and *n* represents the number of observations. CCA finds the projection among the features and the derived equation as:(7)$$X_{\;V_{fused}^{train}}\in {ℛ}^{k},\;X_{\;V_{fused}^{test}\;}\in {ℛ}^{m}$$

The correlation among $X^{T}{}_{\;V_{fused}^{train}}\;P,\;X^{T}{}_{\;V_{fused}^{test}}Q$, where $\;P=\left[p_{1},p_{2},p_{3},\ldots ,p_{n}\right]$, $Q=\left[q_{1},q_{2},q_{3},\ldots ,q_{n}\right]$, denotes the sample vectors. Formally, the CCA is defined as:(8)$$proj_{v}=\max\frac{\left(X^{T}{}_{\;V_{fused}^{train}}\;X_{\;V_{fused}^{test}\;}S_{pq}\right)}{\sqrt{\left(X^{T}{}_{\;V_{fused}^{train}}S_{pp}X_{\;V_{fused}^{train}}\right)\left(X^{T}{}_{\;V_{fused}^{test}}S_{qq}X_{\;V_{fused}^{test}\;}\right)}}$$
where $S_{pq}=PQ^{T}$ presents the covariance matrix among the features sets, and $\;S_{pp}=PP^{T}$ and $S_{qq}=QQ^{T}$ denote the covariance within the two features sets. The resultant matrices are combined using simple concatenation as follows:(9)$$V_{ICCA}\left(i\right)=\left(\begin{array}{c} X^{T}{}_{\;V_{fused}^{train}}P\\ X^{T}{}_{\;V_{fused}^{test}}Q \end{array}\right)$$
where $V_{ICCA}\left(i\right)$ denotes the resultant vector. By this approach, we obtained a fused ICCA-based feature vector of dimension $\times K_{i}$, where i∈{N×1292, N×382, N×2333, N×2049, and N×232}. These *i*th vectors denote the fused vectors of all five selected datasets. These feature vectors were further optimized using an improved tree growth selection algorithm. The working of improved method is described below.

### 3.6. Improved Tree Growth Feature Selection

Feature selection (FS) in computer vision is the process of identifying the most relevant subset of the original group of features by deleting duplicate, unnecessary, or disturbing features [47]. FS enhances the accuracy while accelerating computational time. In this research, we modified the tree growth optimization algorithm for the best feature selection to become improved tree growth optimization. The adopted algorithm draws its inspiration from the battle for food and light among trees [48]. The work of the original tree growth algorithm is described by the following points:
Randomly generate the population of trees using lower and upper bounds, and calculate the fitness values;Considering feature selection, the objective is to minimize the solution. So, the minimum tree value is selected as the fittest value. At the *k^th^* iteration, $P_{GB}^{k}$ presents the global best;Allow local search for *B*1 better solutions using the Equation (10). For every solution, do many local searches. If the value of the new answer is greater than the first response, then replace it with old value.
(10)$$P_{i}^{k+1}=\frac{P_{i}^{k}}{\beta }+rP_{i}^{k}$$
where *β* presents the reduction rate of trees caused by aging high growth, and less food, and *r* presents  U (0,1), which causes the trees to satisfy the light, and the roots are instructed to move, absorb food and grow at a rate of $rP_{i}^{k}$ units.

The best solutions that are close together under different angles should be moved *B*2 solutions to the space between them. Using Equation (11), compute the separation between the selected trees and other trees:(11)$$d_{i}=\sqrt{\left(\overset{B1+B2}{\underset{i=1}{\sum }}{\left(P_{B2}^{j}-P_{i}^{k}\right)}^{2}\right)}\;\&\;d_{i}=\left\{\begin{array}{c} \begin{array}{c} d_{i}\;if\;P_{B2}^{k}\neq P_{i}^{k} \end{array}\\ \infty \;if\;P_{B2}^{k}=P_{i}^{k}\; \end{array}\right\}$$

In the next step, choose two solutions *n*_1_ and *n*_2_ with the smallest *d_i_*, and get a linear combination among the trees. The linear combination is derived in Equation (12).
(12)$$z=\lambda n_{1}+\left(1-\lambda \right)n_{2}$$
where λ=U(0,1) and, finally, the algorithm moves this tree among two adjacent trees with $\theta_{i}=U\;\left(0,\;1\right)$ angles, which are obtained from the Equation (13).
(13)$$P_{B2}^{k}=P_{B2}^{k}+\theta_{i}z$$

*B*3 worse solutions should be replaced with randomly generated ones.

Generate a new population using  B=B1 B2+B3. The mask operator randomly generates *B*4 new solutions, and each new solution is modified in relation to the best solution (from the population *B*1) before being added to the new population (new population = new population + *B*4).

We consider the number of original populations *B* from the new population after sorting it as the starting population for the subsequent iteration.

Compute the entropy value of all populations for each iteration and define an activation function for the selection of important features.

This process continues for the selected number of iterations.

Based on the above selection process, we obtained a feature vector of dimension $\times {\tilde{K}}_{i}$, where i∈{N×637, N×176, N×1177, N×960 and N×97}. These feature vectors were obtained for all selected datasets separately. The final selected features were fed into neural network classifiers for the final classification.

## 4. Experiment and Results

In this section, the experimental approach of the proposed framework is described. This research utilized five datasets for the experimental process (details of the datasets have been described under Section 3.1). The datasets were partitioned in a 50:50 ratio. This indicated that 50% of the sample images were used for training the proposed framework, while the remaining 50% were used for testing. All proposed framework results were evaluated using 10-fold cross validation. Several hyperparameters of deep learning models were initialized statically and by using a Bayesian optimization technique. We established the initial learning rate, stochastic gradient descent, momentum, L2Regularization, dropout rate, activation type, and section depth using a Bayesian optimization method. Neural networks classifiers were utilized to evaluate the results. The evaluating parameters were sensitivity rate, precision rate, false positive rate, kappa index, MCC, accuracy, and time (seconds). The kappa and MCC measures were computed by Equations (14) and (15).
(14)$$k=\frac{P_{0}-P_{e}}{1-P_{e}}$$
where, $P_{e}$ denotes the predicted agreement and $P_{0}$ is the actual agreement. In essence, it reveals you how much better your classifier performs than a classifier that just makes accurate predictions based on the frequency of each class.
(15)$$MCC=\frac{T+ve\times T-ve-F+ve\times F-ve}{\sqrt{\left(T+ve+F+ve\right)\left(T+ve+F-ve\right)\left(T-ve+F+ve\right)\left(T-ve+F-ve\right)}}\;$$
where  T+ve, T−ve, F+ve, F−ve denotes the  True Postive, True Negative, False Positive and False Negative, respectively. We conducted all simulations on MATLAB2022a utilizing a work station PC with an Intel Core i7 5570 processor, 380 SSD, 1TB hard drive, and 32 GB of RAM, as well as a 6 GB NVIDIA RTX graphics card. The results of the proposed framework are presented in two different experiments. In the first experiment, the proposed ICCA-based fusion results are discussed, whereas in the second experiment, the proposed optimization algorithm-based results are discussed.

### 4.1. Chest X-ray (COVID-19 and Pneumonia) Dataset Results

The results of the first experiment (ICCA-based fusion) for the Chest X-ray dataset are presented in Table 3. This table shows that the MNN classifier outperformed the other classifiers. The MNN classifier beat the other classifiers based on the numerical stats. This classifier achieved a 99.6% accuracy rate in 16.316 s. The values for the sensitivity, precision, F1-score, kappa index, MCC rate, and FPR were 99.59, 99.60, 99.60, 99.08, 99.39, and 0.002, respectively. The confusion matrix of the MNN classifier is shown in Figure 6, and it was utilized to verify the MNN classifier accuracy and other computed measures. Computationally, the tri-layered NN required the longest computation time of 29.912 s, whereas the narrow NN classifier required the shortest computation time of 14.949 s.

Table 4 demonstrates the proposed optimization results (second experiment) for the Chest X-ray dataset. This table illustrates that the WNN classifier had the greatest accuracy of 99.6%. For this classifier, the sensitivity was 99.58%, the accuracy was 99.58%, the F1-score was 99.58%, the kappa index was 99.05%, the MCC was 99.37%, and the false positive rate was 0.002. These numbers were also calculated for the other neural kernels, and it was found that the WNN performed the best based on the numerical values. Figure 7 depicts the WNN confusion matrix, which provides further confirmation of the computed values. The computational time was also noted for each classifier, and the shortest time was 9.97 s for the narrow neural network classifier, while the longest time was 12.77 s for the tri-layered neural network. In comparison with the first experiment results of this experiment, it was shown that the accuracy was consistent for the initial three selected classifiers, but for the last two classifiers, the accuracy was improved. Also, the computational time was significantly reduced, which was the strength of this experiment.

### 4.2. COVID-19 Patients Lungs X-ray Images Dataset

Table 5 presents the classification results of the proposed features fusion for the COVID-19 Patients Lungs X-ray dataset. In this table, the WNN classifier achieved the maximum accuracy of 98.5%. The sensitivity rate for this classifier was 98.56%, the F1-score was 98.48%, the kappa index was 97.0%, the MCC rate was 97.03%, and the FPR was 0.002. These measures were also computed for the rest of the classifiers mentioned in this table. According to these results, it was noted that the WNN classifier achieved the better accuracy. Figure 8 illustrates the confusion matrix of the WNN classifier for further validation of the statistical values. In addition, the computational time was noted for each classifier during the testing stage, and it was observed that the bi-layered neural network took the minimum time for execution at 2.594 s.

Table 6 presents the proposed optimization results on this dataset and the obtained improved accuracy of 98.5% for the WNN. The other measures were a sensitivity rate of 98.56%, a precision rate of 98.34%, an F1-score of 98.48%, a kappa index of 97.0%, an MCC of 97.03%, and an FPR of 0.002. The computational time of this experiment was also improved from the first experiment. Moreover, the Figure 9 showing the confusion matrix of WNN for this dataset. Hence, we can conclude that the optimization process improved the performance.

### 4.3. COVID-19 Lung CT Scans Dataset Results

Table 7 illustrates the proposed fusion results for the Covid-19 Lung CT Scans dataset. This table demonstrates that the WNN classifier achieved the highest accuracy of 99.3% across all neural networks. The other computed measures of the WNN were a sensitivity rate of 99.25%, a precision rate of 98.9%, an F1-score of 99.23%, a kappa index of 98.12%, an MCC rate of 98.12%, and an FPR of 0.01. The same computations were also conducted on all other neural network classifiers, and it was seen that the WNN classifier performed better. Figure 10 depicts the confusion matrix of the WNN that was used to validate the obtained values (i.e.*,* accuracy, precision). Computational time was also noted for all classifiers, and the MNN classifier had the minimum time of 30.96 s, while the WNN classifier consumed the maximum time of 41.36 s.

Table 8 describes the proposed optimization results for the COVID-19 Lung CT Scans dataset. This table shows that the narrow NN classifier performed better with a lower computational time of 14.747 s and a maximum accuracy of 99.9%, which was better than the other mentioned classifiers. The sensitivity rate was 99.93%, the precision rate was 99.81%, the F1-score rate was 99.88%, the kappa index was 99.70%, the mean correlation coefficient (MCC) was 99.70%, and the FPR value was 0.001. On the basis of these statistics, it was affirmed that that the NNN classifier was superior to the other stated neural networks. Figure 11 shows the confusion matrix of the NNN that was utilized to verify the obtained values. The execution time of the other neural networks was also noted, and the medium NN classifier took the minimum time of 14.687 s, whereas the maximum recorded time was 19.961 s for the WNN. In comparison with the first experiment results, it was noted that the performance was improved for the second experiment. Moreover, it was also clearly observed that the computational time was reduced for the second experiment than for the first experiment. Hence, the second experiment showed the main strength in terms of accuracy and time for this dataset.

### 4.4. COVID-19 Image Dataset Results

The results of this dataset using the proposed fusion method (first experiment) are presented in Table 9. This table demonstrates that the maximum accuracy came from the MNN classifier at 97.2%. The values of other measures of this classifier include sensitivity rate, accuracy rate, F1-Score, Kappa index, MCC, and FPR values of 99.88%, 96.94%, 96.90%, 97.30%, 96.23%, and 0.006, respectively. These values were also computed for the other neural classifiers, and the numerical results demonstrated that MNN outperformed the other classifiers. Figure 12 depicts a confusion matrix that was used to validate these MNN classifier performance values. The computational time was also noted for this experiment, and it was observed that the minimum computed time was 25.885 s for the medium NN, whereas the maximum time was 89.762 s for the bi-layered NN.

Table 10 presents the optimization results (second experiment) that show the maximum achieved accuracy was 99.5% for the MNN classifier. Moreover, the sensitivity rate was 99.43%, the F1-Score was 99.45%, the kappa index value was 98.45%, and the MCC was 99.33%. These values were also computed for the other classifiers shown in this table. Based on these values, it was shown that the MNN performance was better. The values obtained by the MNN could be further verified through a confusion matrix, as is shown in Figure 13. Moreover, computationally, the narrow neural classifier took the least amount of time at 15.298 s. In comparison of this dataset result among both experiments, we conclude that the optimization accuracy was better than in the first experiment. Also, the computational time was significantly reduced in the second experiment.

### 4.5. COVID-19 Detection Dataset Results

Table 11 presents the results of this dataset for the first experiment. In this table, it is shown that the MNN classifier had highest accuracy of 100%, which was verified through a confusion matrix that is illustrated in Figure 14. Computationally, the narrow neural network classifier had the best time of 2.1373 s. Moreover, the other classifiers also obtained the maximum accuracies up to 99.9%. Table 12 presents the classification accuracy of the second experiment (optimization) that shows the maximum accuracy of 100%. The accuracy of both experiments was almost consistent, but the time was changed. The computational time of the second experiment was lower than in the first experiment, and 1.4186 s was the minimum reported time. Moreover, Figure 14 and Figure 15 shows the confusion matrix of MNN and tri-layered NN for this dataset. 

In conclusion, a comprehensive comparison was conducted with several recent techniques, as is shown in Table 13. Several latest released techniques are included in this table, and they all used the deep learning architecture. Recently, the highest recorded accuracy was 99.2%. Our framework tested on five publically available datasets achieved accuracies of 99.6, 98.5, 99.9, 99.5, and 100%.

A comprehensive comparison was conducted with several recent techniques, as is shown in Table 13. Several latest released techniques are included in this table, and they all used the deep learning architecture. Recently, the highest recorded accuracy was 99.2%. Our framework tested on five publically available datasets achieved accuracies of 99.6, 98.5, 99.9, 99.5 and 100%. Moreover, the in-depth analysis among original CCA and ICCA was conducted at the end to show the change in accuracy after the improved fusion method. Table 14 shows the results of this process, and it was observed that the accuracy of the ICCA was improved, but the time also increased slightly. Overall, the ICCA-based fusion approach showed the most improvement. Moreover, Figure 16 shows the impact of the data augmentation process. This figure illustrates that the accuracy of the proposed framework was improved after the data augmentation. In addition, Figure 17 illustrates the infected region visualization using the Grad-CAM approach on the proposed framework.

## 5. Conclusions

In this work, we proposed a framework for COVID-19 classification using deep Bayesian optimization and an improved feature selection algorithm. The proposed framework started with the augmentation process. This step improved the accuracy of the proposed framework when it was compared with the original dataset results. The training of deep models was performed through transfer learning, whereas the hyperparameters were initialized through Bayesian optimization instead of manual initialization. The manual initialization process did not improve the learning capability of selected deep models, as determined after the experimentation process. After that, the deep features were extracted and fused using an ICCA-based approach instead of serial-based fusion. This process not only increased the accuracy, but it also controlled the computational time. The accuracy and time of the original CCA-based approach was not better than the ICCA-based fusion. Based on the analysis, it was also observed that a few redundant features were also present during the ICCA-based fusion, which was a drawback of this step; therefore, we proposed an improved optimization algorithm based on an entropy activation function. This step improved the accuracy and reduced the testing computational time. The experimental process was conducted on five publically available datasets and obtained improved accuracies of 99.6, 98.5, 99.9, 99.5, and 100%. The main drawback of this work was the use of complex dataset and fusion processes that also added redundant information. Moreover, the ICCA-based fusion method added some redundant information that caused higher computational times. These drawbacks will be considered in a future work [54,55].

## Figures and Tables

**Figure 1 diagnostics-13-00101-f001:**
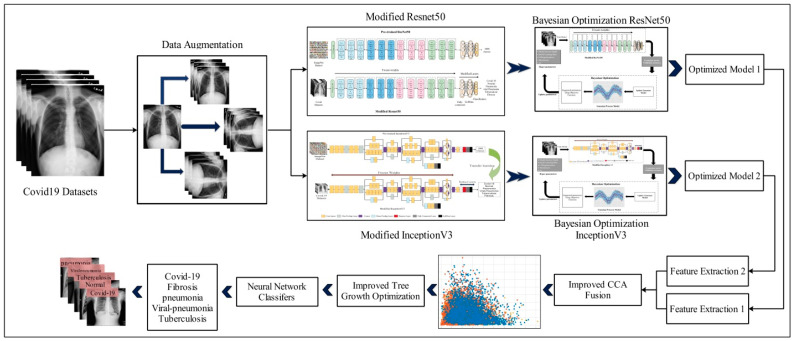
Proposed methodology for diagnosing COVID-19 using Bayesian optimization and improved tree growth optimization.

**Figure 2 diagnostics-13-00101-f002:**
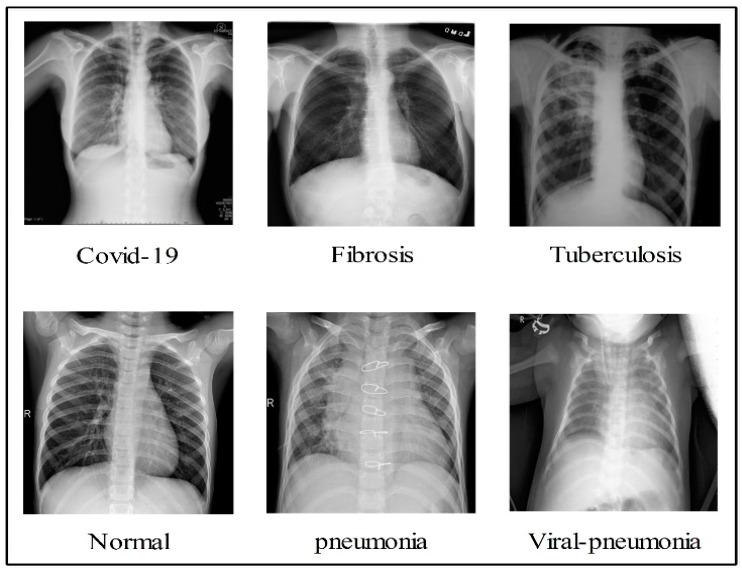
Sample images of each class infections.

**Figure 3 diagnostics-13-00101-f003:**
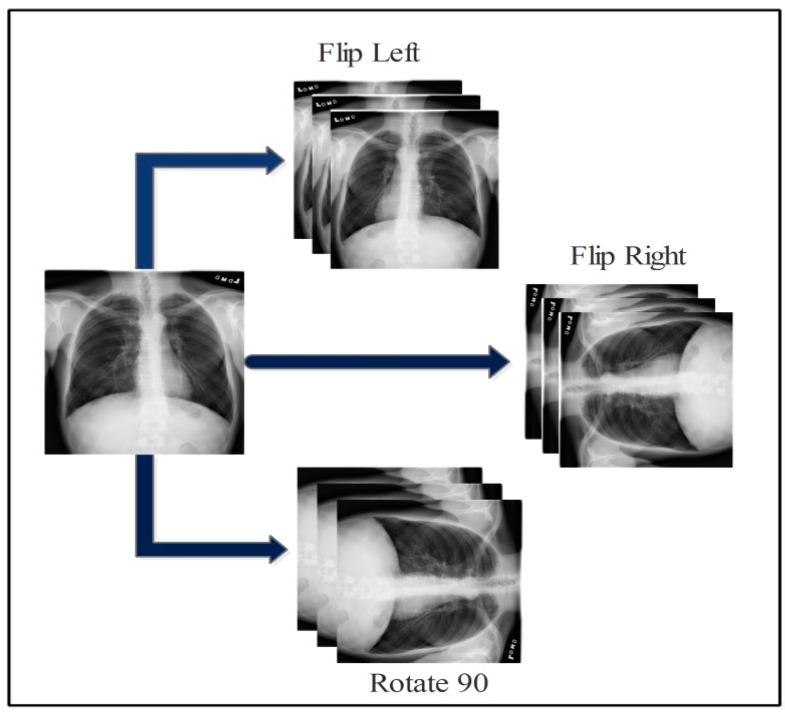
Visual presentation of augmentation process for COVID-19 datasets.

**Figure 4 diagnostics-13-00101-f004:**
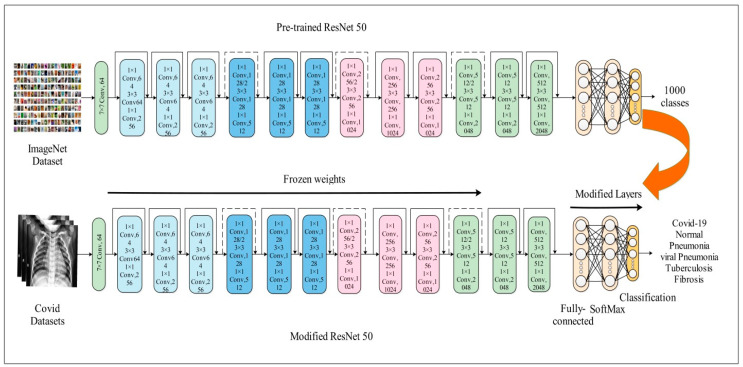
Modified architecture of ResNet50 using freezing weights.

**Figure 5 diagnostics-13-00101-f005:**
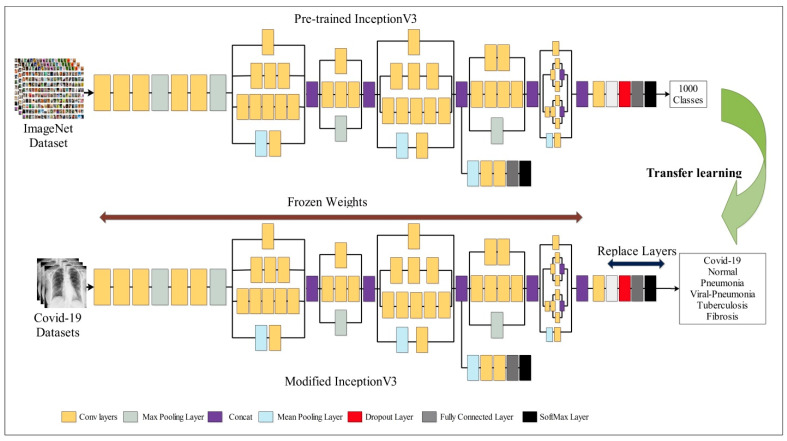
Updated architecture of Inception v3 using freezing weights via transfer learning.

**Figure 6 diagnostics-13-00101-f006:**
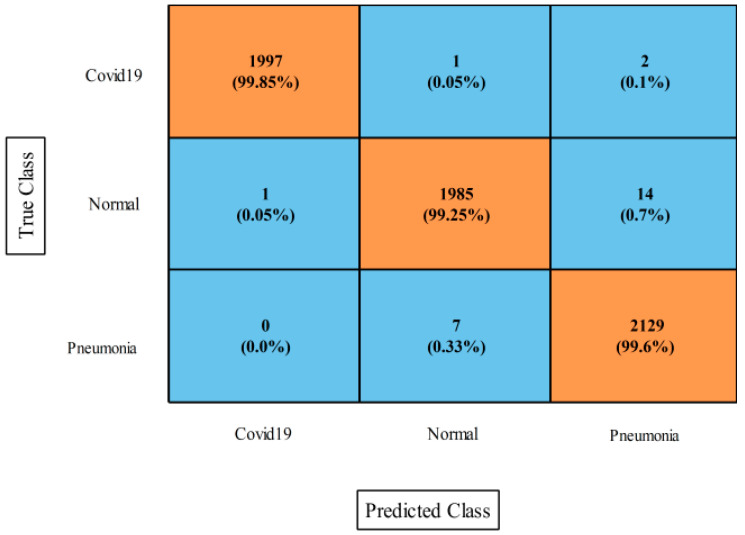
Confusion matrix of MNN for proposed fusion method on Chest X-ray dataset.

**Figure 7 diagnostics-13-00101-f007:**
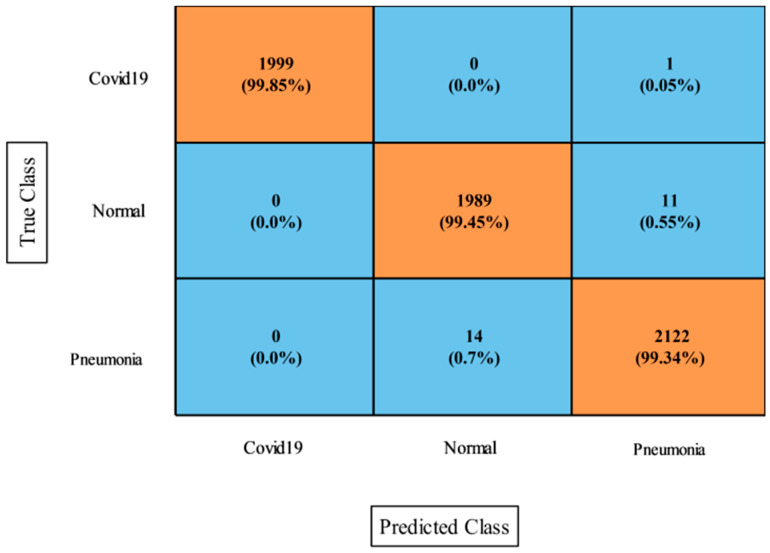
Confusion matrix WNN for proposed improved tree growth optimization method on Chest X-ray dataset.

**Figure 8 diagnostics-13-00101-f008:**
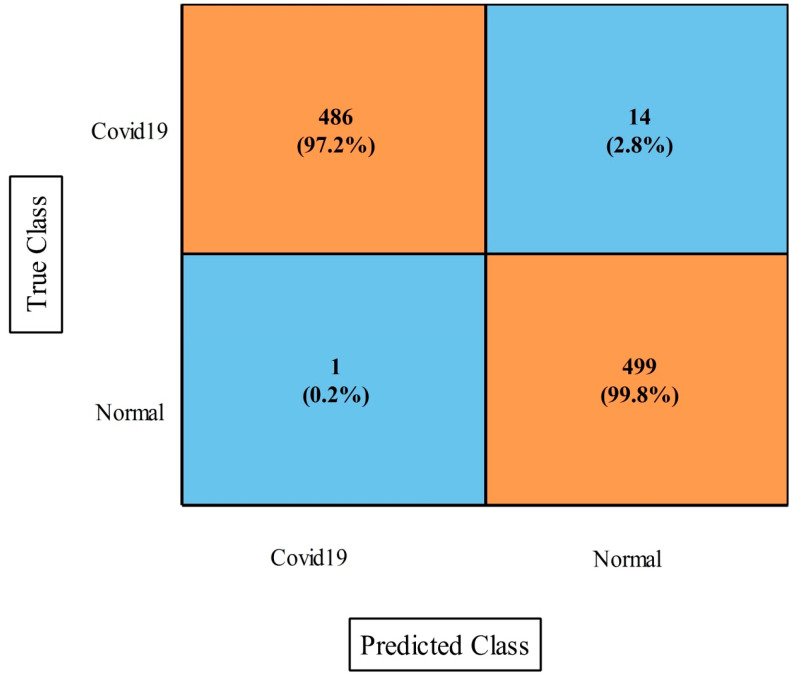
Proposed fusion approach confusion matrix for WNN classifier on COVID-19 Patients Lungs X-ray dataset.

**Figure 9 diagnostics-13-00101-f009:**
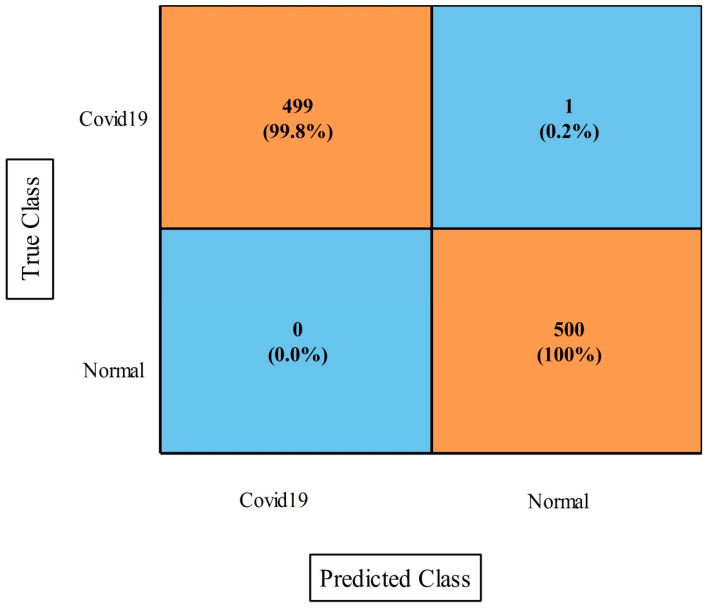
Proposed optimization confusion matrix of WNN classifier on COVID-19 Patients Lungs X-ray dataset.

**Figure 10 diagnostics-13-00101-f010:**
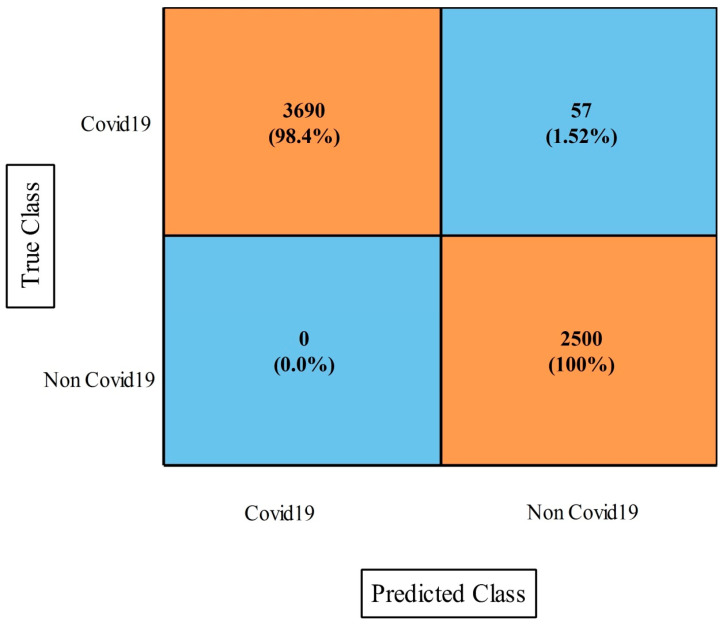
Proposed fusion method confusion matrix of WNN on COVID-19 Lung CT Scans dataset.

**Figure 11 diagnostics-13-00101-f011:**
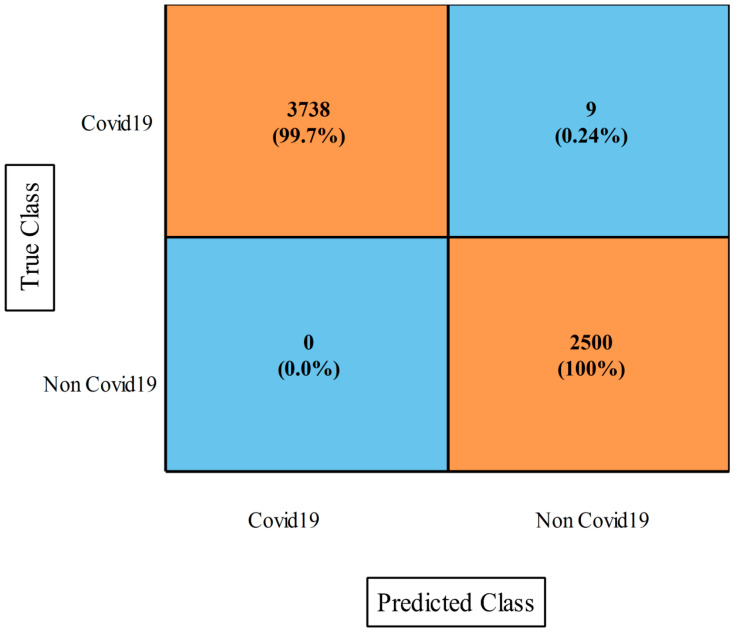
Confusion matrix of NNN classifier using proposed optimization on COVID-19 Lung CT Scans dataset.

**Figure 12 diagnostics-13-00101-f012:**
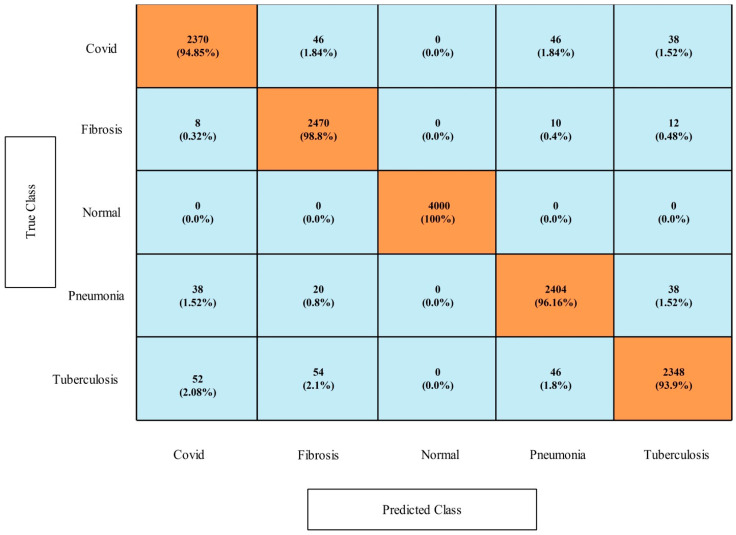
Proposed improved CCA fusion method confusion matrix of MNN on COVID-19 Image dataset.

**Figure 13 diagnostics-13-00101-f013:**
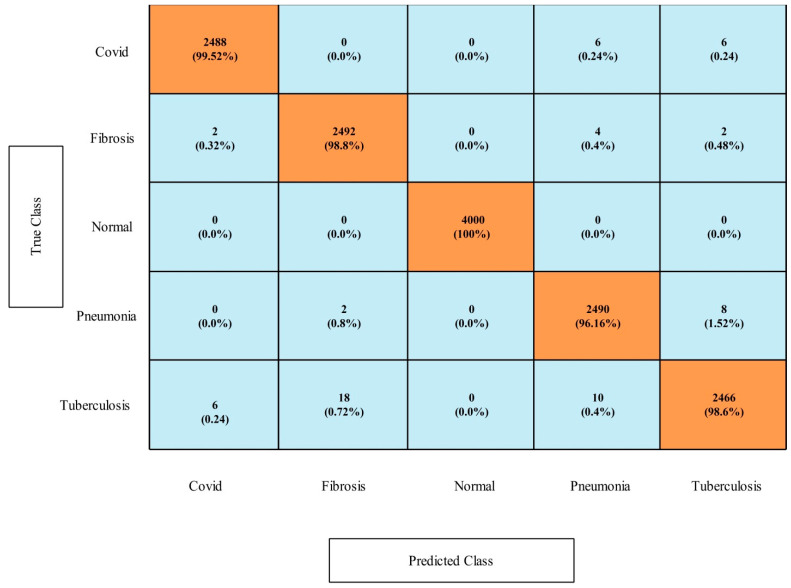
Proposed optimization approach confusion matrix of MNN classifier for COVID-19 Image dataset.

**Figure 14 diagnostics-13-00101-f014:**
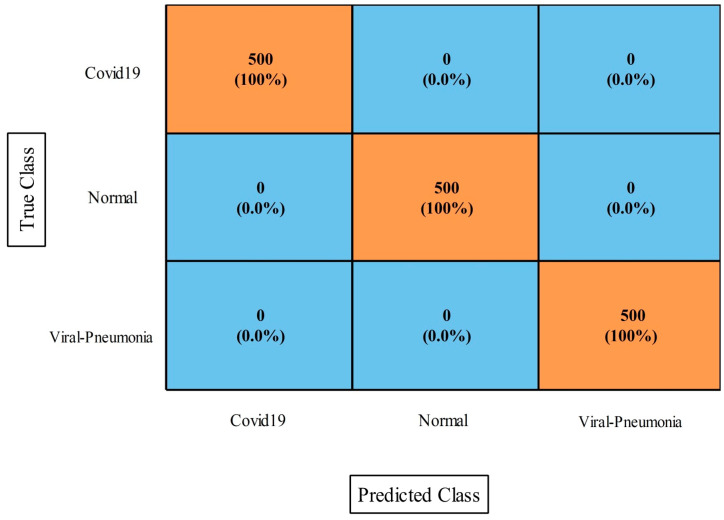
Confusion matrix of MNN learner on COVID-19 Detection dataset.

**Figure 15 diagnostics-13-00101-f015:**
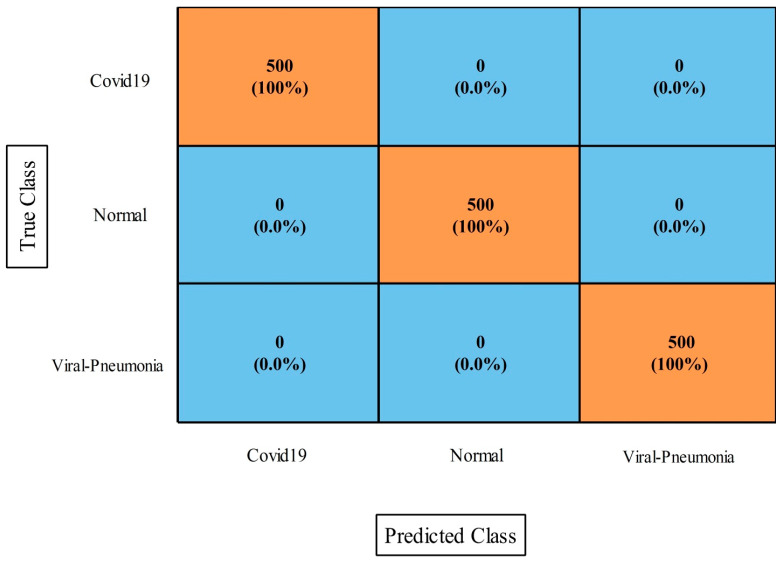
Confusion matrix of tri-layered neural network classifier on COVID-19 Detection dataset.

**Figure 16 diagnostics-13-00101-f016:**
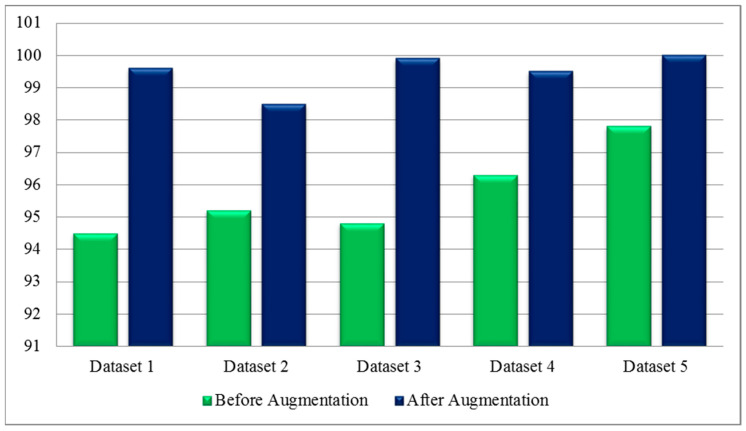
Classification results before and after data augmentation.

**Figure 17 diagnostics-13-00101-f017:**
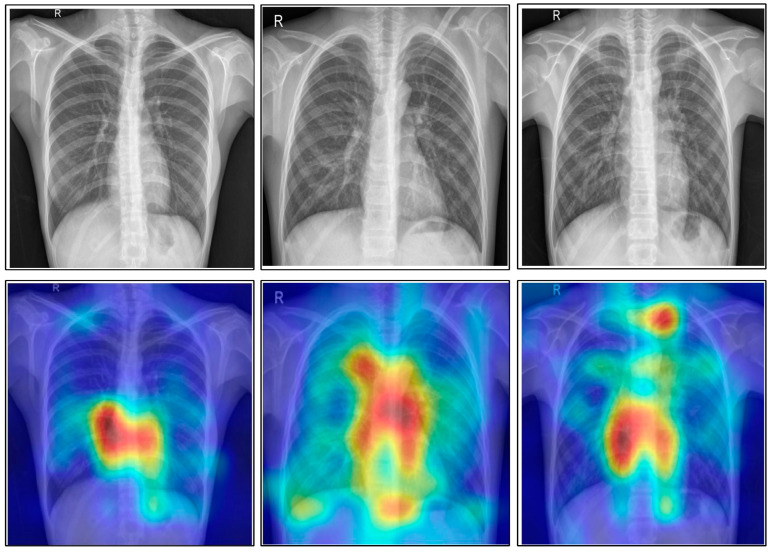
Grad-CAM-based infected region visualization.

**Table 1 diagnostics-13-00101-t001:** Description of five different COVID-19 datasets.

Dataset	Training Images	Augmented Images	Training/Testing Images
**Chest X-ray (COVID-19 & Pneumonia)** COVID-19 Normal Pneumonia			
	576 1583 4273	4000 4000 4273	2000/2000 2000/2000 2137/2136
**COVID19 Patients Lungs X-ray Images** COVID-19 Normal			
	70 28	1000 1000	500/500 500/500
**COVID-19 Lung CT Scans** COVID NON-COVID			
	7496 944	7495 5000	3748/3747 2500/2500
**COVID-19 Detection** COVID-19 Fibrosis Normal Pneumonia Tuberculosis			
	3616 1686 11,767 4265 3500	5000 5000 8000 5000 5000	2500/2500 2500/2500 4000/4000 2500/2500 2500/2500
**COVID-19 Image Dataset** COVID-19 Normal Viral Pneumonia			
	111 70 70	1000 1000 1000	500/500 500/500 500/500

**Table 2 diagnostics-13-00101-t002:** Ranges of hyperparameters for BO.

Hyperparameters	Ranges
Learning Rate	[0.0001, 1]
Section Depth	[1, 3]
Momentum	[0.5, 0.98]
L2Regularization	[1*e*^−9^, 1*e*^−3^]
Dropout	[0.0, 0.8]
Activation type	RELU, tanh, sigmoid

**Table 3 diagnostics-13-00101-t003:** Proposed improved CCA fusion results on Chest X-ray dataset.

Classifiers	Precision	Sensitivity	F1-Score	FPR	Kappa	MCC	Accuracy	Time (s)
NNN	99.54	99.53	99.53	0.002	98.94	99.29	99.5	14.949
**MNN**	**99.60**	**99.59**	**99.60**	**0.002**	**99.08**	**99.39**	**99.6**	**16.316**
WNN	99.59	99.58	99.58	0.002	99.05	99.37	99.6	21.932
Bi-layered NN	98.90	98.89	98.90	0.005	97.51	98.34	98.9	19.737
Tri-layered NN	98.57	98.57	98.57	0.007	96.67	97.85	98.6	29.912

**Table 4 diagnostics-13-00101-t004:** Proposed improved TGO optimization results on Chest X-ray dataset.

Classifiers	Precision	Sensitivity	F1-Score	FPR	Kappa	MCC	Accuracy	Time (s)
NNN	99.50	99.50	99.50	0.002	98.86	99.25	99.5	9.9754
MNN	99.55	99.55	99.55	0.002	98.97	99.32	99.5	8.6999
**WNN**	**99.58**	**99.58**	**99.58**	**0.002**	**99.05**	**99.37**	**99.6**	**11.157**
Bi-layered NN	99.21	99.21	99.21	0.004	98.20	98.81	99.2	12.029
Tri-layered NN	99.27	99.21	99.21	0.004	98.20	98.81	99.1	12.775

**Table 5 diagnostics-13-00101-t005:** Proposed fusion results on COVID-19 Patients Lungs X-ray dataset.

Classifiers	Precision	Sensitivity	F1-Score	FPR	Kappa	MCC	Accuracy	Time (s)
NNN	97.05	96.94	96.81	0.03	93.80	93.96	96.9	3.2829
MNN	98.21	98.32	98.27	0.002	96.60	96.64	98.3	3.1172
**WNN**	**98.34**	**98.56**	**98.48**	**0.002**	**97.00**	**97.03**	**98.5**	**2.9841**
Bi-layered NN	96.15	96.00	95.88	0.04	92.00	92.17	96.0	2.5945
Tri-layered NN	95.36	95.14	94.92	0.04	90.20	90.42	95.1	2.9175

**Table 6 diagnostics-13-00101-t006:** Proposed feature optimization results on COVID-19 Patients Lungs X-ray dataset.

Classifiers	Precision	Sensitivity	F1-Score	FPR	Kappa	MCC	Accuracy	Time (s)
NNN	97.05	96.94	96.81	0.03	93.80	93.96	96.9	3.2829
MNN	98.21	98.32	98.27	0.002	96.60	96.64	98.3	3.1172
**WNN**	**98.34**	**98.56**	**98.48**	**0.002**	**97.00**	**97.03**	**98.5**	**2.9841**
Bi-layered NN	96.15	96.00	95.88	0.04	92.00	92.17	96.0	2.5945
Tri-layered NN	95.36	95.14	94.92	0.04	90.20	90.42	95.1	2.9175

**Table 7 diagnostics-13-00101-t007:** Proposed fusion results on COVID-19 Lung CT Scans dataset.

Classifiers	Precision	Sensitivity	F1-Score	FPR	Kappa	MCC	Accuracy	Time (s)
NNN	98.55	99.00	99.00	0.01	97.54	97.57	98.8	31.86
MNN	98.62	99.00	99.03	0.01	97.61	97.63	98.8	30.96
**WNN**	**98.9**	**99.25**	**99.23**	**0.01**	**98.11**	**98.12**	**99.3**	**41.36**
Bi-layered NN	95.90	96.94	96.45	0.035	95.39	95.39	96.4	37.27
Tri-layered NN	95.65	94.55	95.75	0.04	91.66	91.63	96.0	41.88

**Table 8 diagnostics-13-00101-t008:** Proposed ITCO optimization results on COVID-19 Lung CT Scans dataset.

Classifiers	Precision	Sensitivity	F1-Score	FPR	Kappa	MCC	Accuracy	Time (s)
**NNN**	**99.81**	**99.93**	**99.88**	**0.001**	**99.70**	**99.70**	**99.9**	**14.747**
MNN	99.85	99.75	99.83	0.001	99.57	99.57	99.8	14.687
WNN	99.73	99.82	99.85	0.001	99.70	99.70	99.9	19.961
Bi-layered NN	98.62	98.15	98.95	0.005	97.40	97.42	98.4	19.054
Tri-layered NN	96.95	97.05	97.56	0.032	93.94	93.94	97.1	37.34

**Table 9 diagnostics-13-00101-t009:** Proposed fusion method results on COVID-19 Image dataset.

Classifiers	Precision	Sensitivity	F1-Score	FPR	Kappa	MCC	Accuracy	Time
NNN	95.72	95.68	95.69	0.009	87.95	94.75	96.1	42.351
**MNN**	**96.94**	**96.88**	**96.90**	**0.006**	**97.30**	**96.23**	**97.2**	**25.885**
WNN	96.66	96.66	96.65	0.007	90.67	95.93	97.0	36.578
Bi-layered NN	92.22	92.18	92.19	0.017	78.17	90.49	93.0	89.762
Tri-layered NN	91.53	91.46	91.49	0.018	76.16	89.63	92.4	87.431

**Table 10 diagnostics-13-00101-t010:** Proposed ITGO optimization approach results on COVID-19 Image dataset.

Classifiers	Precision	Sensitivity	F1-Score	FPR	Kappa	MCC	Accuracy	Time
NNN	99.04	99.02	99.03	0.002	97.28	98.82	99.1	15.298
**MNN**	**99.47**	**99.43**	**99.45**	**0.002**	**98.45**	**99.33**	**99.5**	**12.939**
WNN	99.49	99.49	99.49	0.001	98.57	99.38	99.5	20.814
Bi-layered NN	97.42	97.41	97.41	0.005	92.77	96.85	97.7	19.818
Tri-layered NN	97.79	97.79	97.79	0.004	93.84	97.31	98.0	45.819

**Table 11 diagnostics-13-00101-t011:** Proposed ICCA fusion approach results on COVID-19 Detection dataset.

Classifiers	Precision	Sensitivity	F1-Score	FPR	Kappa	MCC	Accuracy	Time
NNN	99.93	99.93	99.93	0.00	99.85	99.90	99.9	2.1373
**MNN**	**100**	**100**	**100**	**0.00**	**100**	**100**	**100**	**2.2438**
WNN	100	100	100	0.00	100	100	100	2.8256
Bi-layered NN	99.93	99.93	99.93	0.00	99.85	99.90	99.9	2.1971
Tri-layered NN	99.93	99.93	99.93	0.00	99.85	99.90	99.9	2.4073

**Table 12 diagnostics-13-00101-t012:** Proposed ICCA fusion approach results on COVID-19 Detection dataset.

Classifiers	Precision	Sensitivity	F1-Score	FPR	Kappa	MCC	Accuracy	Time
NNN	100	100	100	0.00	100	100	100	2.3249
MNN	100	100	100	0.00	100	100	100	1.97
WNN	100	100	100	0.00	100	100	100	2.0038
Bi-layered NN	99.93	99.93	99.93	0.00	99.85	99.90	99.9	1.4186
Tri-layered NN	**100**	**100**	**100**	**0.00**	**100**	**100**	**100**	**1.5376**

**Table 13 diagnostics-13-00101-t013:** Proposed approach comparison with latest techniques.

Sr. No	Reference	Year	Method	Accuracy (%)
**1**	[49]	2022	Quantum Machine learning using GAN	95.0
**2**	[50]	2022	Classical and quantum transfer learning for COVID-19 classification	99.0
**3**	[51]	2022	Neighboring aware based deep graph network	99.2
**4**	[52]	2021	Ensemble Convolutional neural network	97.0
**5**	[53]	2021	Deep based fusion transfer learning and DCA	98.3
**Proposed** Deep learning, Bayesian optimization, ICCA fusion and best features selection				**99.6** **98.5** **99.9** **99.5** **100**

**Table 14 diagnostics-13-00101-t014:** Analysis of original CCA-based features fusion and ICCA-based fusion.

Datasets and Classifier	CCA	ICCA	Accuracy (%)	Time (s)
Chest X-ray (COVID-19 and Pneumonia)	✓		98.4	14.215
		✓	99.6	16.316
COVID-19 Patients Lungs X-ray Images	✓		96.2	1.9624
		✓	98.5	2.9841
COVID-19 Lung CT Scans	✓		98.9	37.142
		✓	99.3	41.368
COVID-19 Detection	✓		96.0	22.204
		✓	97.2	25.885
COVID-19 Image Dataset	✓		99.2	2.0426
		✓	100	2.2438

## Data Availability

The datasets used in this work are publically available. The selected datasets are Chest X-ray (https://www.kaggle.com/datasets/prashant268/chest-xray-covid19-pneumonia (accessed on 28 November 2022)), COVID-19 Patients Lungs X-ray Images (https://www.kaggle.com/datasets/nabeelsajid917/covid-19-x-ray-10000-images (accessed on 28 November 2022)), COVID-19 Lung CT Scans (https://www.kaggle.com/datasets/luisblanche/covidct (accessed on 28 November 2022)), COVID-19 Detection (https://www.kaggle.com/datasets/donjon00/covid19-detection (accessed on 28 November 2022)), and COVID-19 Image Dataset (https://www.kaggle.com/datasets/pranavraikokte/covid19-image-dataset (accessed on 28 November 2022)).

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
