# Peer review of "D2BOF-COVIDNet: A Framework of Deep Bayesian Optimization and Fusion-Assisted Optimal Deep Features for COVID-19 Classification Using Chest X-ray and MRI Scans"

_diagnostics, 2022, doi:10.3390/diagnostics13010101_

Round 1

Reviewer 1 Report

The authors used Deep Bayesian Optimization and Fusion Assisted Optimal Deep Features for COVID-19 Classification. They claim this method          to be unique for X-ray and MRI scans.

I have some suspicion about the unique contribution because I have found some papers using similar methods in the literature.

Compared with other published material, the work adds to the subject area a new method for COVID-19 classification.

The authors have to justify the fact that the method is an improvement over other people who have used the similar method.

Some important references are missing. I have only identified some of them. They must explore all the major ref. in this area.

Following articles need to be used in the literature and for finding the research gap.

1. Khan, I.U.; Aslam, N. A Deep-Learning-Based Framework for Automated Diagnosis of COVID-19 Using X-ray Images. Information 202011, 419. https://doi.org/10.3390/info11090419

2. Ullah, N.; Khan, J.A.; Almakdi, S.; Khan, M.S.; Alshehri, M.; Alboaneen, D.; Raza, A. A Novel CovidDetNet Deep Learning Model for Effective COVID-19 Infection Detection Using Chest Radiograph Images. Appl. Sci. 2022, 12, 6269. https://doi.org/10.3390/ app12126269

3. Shifat E Arman, Sejuti Rahman, Shamim Ahmed Deowan et al. COVIDXception-Net: A Bayesian Optimization Based Deep Learning Approach to Diagnose COVID-19 from X-Ray Images, 18 August 2020, PREPRINT (Version 1) available at Research Square [https://doi.org/10.21203/rs.3.rs-58531/v1]

I am not quite clear whether your work is an improvement over these articles. This is the main limitation of this paper.

Author Response

Response sheet attached. thanks

Reviewer 2 Report

Diagnostics | An Open Access Journal from MDPI

The following is an overview of the article D2BOF-COVIDNet: A Framework of Deep Bayesian Optimization and Fusion Assisted Optimal Deep Features for COVID-19 Classification using Chest X-ray and MRI Scans (Manuscript ID diagnostics-2111951). In the proposed framework, initially performed data augmentation for better training of the selected deep models. After that, two pretrained deep models are employed (ResNet50 and InceptionV3) and trained using transfer learning.

The author(s) stated in the first part of the study; In December of 2019, consumption of bat meat at an unusual animal meat market in Wuhan, Hubei, China was connected to a group of people having pneumonia of unknown cause. The pandemic soon spread to other regions of the globe, and on March 11, 2020, the World Health Organization declared COVID-19 a worldwide pandemic outbreak that is continuing. A new type of beta coronavirus was found using unbiased sequencing of patient samples. Comparing the 2019-nCoV coronavirus to Middle East Respiratory Syndrome (MERS) and severe acute respiratory syndrome (SARS), it has a higher transmission potential and a lower mortality rate SARS and MERS are both animal-borne diseases and civets and camels, respectively, were known to carry diseases. The emergence of  diseases like SARS and MERS, both of which are suspected to have been brought on by new coronaviruses, is more possible when there are no borders between human civilization and the natural environment. In the early days, continuing the number of deaths because of no vaccination. The manual diagnosis of this disease using Chest X-Ray (CXR) images and MRI is time consuming and always need an expert person. Therefore, a computerized technique is widely required that can be helpful for the radiologists as a second opinion. The recent computerized techniques face some challenges such as low contrast CTX images, manual initialization of hyperparameters and redundant features. In this  paper, we proposed a novel framework for COVID-19 classification using deep Bayesian optimization and improved canonical correlation analysis (ICCA). In the proposed framework, initially we performed data augmentation for better training of the selected deep models. After that, two pretrained deep models are employed (ResNet50 and InceptionV3) and trained using transfer learning. The hyperparameters of both models are initialized through Bayesian optimization. Both trained  models are utilized for the features extraction and fused using an ICCA based approach. The fused features are further optimized using an improved tree growth optimization algorithm that finally classified using neural network classifier. The experimental process is conducted on five publically available datasets and achieved an accuracy of 99.6, 98.5, 99.9, 99.5, and 100%. The comparison with  recent methods and t-test based analysis shows the significance of proposed framework.

The author(s) stated in the last part of the study; in this work, we proposed a framework for COVID-19 classification using deepBayesian optimization and improved optimization algorithm. The proposed framework starts with the augmentation process in which increased the dataset size to improve the  learning capability of fine-tuned deep models. The training of deep models was performed through transfer learning, whereas the hyperparameters was initialized through Bayesian Optimization. The manual initialization process not improves the learning capability of selected deep models. After that, the deep features are extracted and fused using ICCA based approach instead of serial based fusion. This process not only increases the accuracy but also control the computational time. Based on the analysis, it is also observed that a few redundant features are also present; therefore, we proposed an improved optimization algorithm based on entropy activation function. This step improves the accuracy and reduced the testing computational time. The experimental process was conducted on  five publically available datasets and obtained improved accuracy. The main drawback of this work was the use of complex dataset and fusion process that also added the redundant information. This drawback will be considered as a future work.

However, some points must be highlighted so that the author(s) can review and submit in another round of review: The following corrections are considered to be beneficial for the strengthening of the article.

1. The Conclusions should be reviewed again. The original aspect of the study and its difference from other studies should be clearly explained. (The conclusion should be explored better and it needs to contemplate the eventual restrictions of the developed technique to address future works in this area.)

2. The abstract must be make strong. Abstract should be reviewed again.

3. Some sentences have spelling errors. (Punctuation marks, spaces, etc.). Some places should be left space.

4. It has been a comprehensive study in the literature in recent years (Especially Swarm Intelligence algorithms, Image segmentation, COVID-19 CT-Scan Images). If there are more current literature studies, these should be examined in detail and added to the literature section. It is a suggestion for the literature part of the article to be more comprehensive. It may be useful to include relevant articles in 2018-2022 in references. As an example, I think it might be useful to add the article to references, such as the articles below, to keep the article updated as a literature.

(1) A new approach to COVID-19 detection from X-ray images using angle transformation with GoogleNet and LSTM. Measurement Science and Technology. 2) A topic-based hierarchical publish/subscribe messaging middleware for COVID-19 detection in X-ray image and its metadata. Soft Computing, 1-11. 3) Diagnosing COVID-19 from X-Ray images with using multi-channel CNN architecture, https://doi.org/10.17341/gazimmfd.746883. 4) Deep learning-based COVID-19 detection system using pulmonary CT scans. Turkish Journal of Electrical Engineering and Computer Sciences, 29(8), 2716-2727.)

5. The authors should compare the results of their method with those of previous studies. As mentioned in the literature, there are several methods with very high accuracy, even better than the proposed method. Author(s) can do compare table (A new table can add about previous studies to result section.). This subject is very important.

6. The motivations of the proposed method are not clear. Which problem does the proposed method attempt to solve? Why the other existing diagnosis methods failed to solve it? What are the advantages of the proposed method compared to other methods? Those should be illustrated more clearly.

7. Carefully check all grammatical error. Still, the English language should be improved. I suggest asking for help from a native English

I think it is ACCEPTABLE after the MAJOR Revisions mentioned. (Reconsider after major revision.)

Author Response

Response Sheet attached. thanks

Reviewer 3 Report

In this manuscript, the authors combine Deep Learning algorithms, Bayesian optimization, Feature Fusion and Feature Selection in order to develop several models for classification of Chest X-ray and MRI images. The topic is interesting and worth investigating. The manuscript is well-structured and well-written. The proposed methodology is validated using five publicly available datasets of COVID-19 lung scans. The obtained results show the effectiveness of the authors’ methodology.

My remarks are as follows:

In the abstract and Introduction sections, please highlight your contributions.

Please add a short description of the paper’s structure at the end of the “Introduction” section.

The comparison of time complexity (Table 14) should be based on an analytical expression.

Technical remarks:

Some abbreviations are not described: MRI, DL, TL…

l. 48, l. 52: The phrases should be edited.

Figure 4: The font size should be enlarged.

Author Response

Response sheet attached. thanks

Round 2

Reviewer 1 Report

Good.

Reviewer 2 Report

Diagnostics | An Open Access Journal from MDPI

Dear Editor;

The author(s) made all the corrections mentioned (Diagnostics-2111951.R1 - D2BOF-COVIDNet: A Framework of Deep Bayesian Optimization and Fusion Assisted Optimal Deep Features for COVID-19 Classification using Chest X-ray and MRI Scans).

The length of the paper is enough in terms of a scientific paper. Considering studies conducted and results obtained, it is believed that the paper is eligible to be published in your journal after your approval.

I think it is ACCEPTABLE  in your journal after your approval as editor.
